



**Development of an integrated socio-hydrological modeling framework**
**for assessing the impacts of shelter location arrangement and human**
**behaviors on flood evacuation processes**
Erhu Du[1], Feng Wu[2], Hao Jiang[3], Naliang Guo[2], Yong Tian[3], and Chunmiao Zheng[3,4*]
[1]Yangtze Institute for Conservation and Development, Hohai University, Nanjing, China
[2]Key Laboratory of Land Surface Pattern and Simulation, Institute of Geographic Sciences
and Natural Resources Research, Chinese Academy of Sciences, Beijing, China
[3]State Environmental Protection Key Laboratory of Integrated Surface Water-Groundwater
Pollution Control, School of Environmental Science and Engineering, Southern University
of Science and Technology, Shenzhen, China
[4]EIT Institute for Advanced Study, Ningbo, Zhejiang, China
*Corresponding to: Chunmiao Zheng, zhengcm@sustech.edu.cn
**Abstract**
In many flood-prone areas, it is essential for emergency responders to use advanced
computer models to assess flood risk and develop informed flood evacuation plans.
However, previous studies have limited understanding of how evacuation performances
are affected by the arrangement of evacuation shelters regarding their number and
geographical distribution and human behaviors regarding the heterogeneity of household
evacuation preparation times and route searching strategies. In this study, we develop an
integrated socio-hydrological modeling framework that couples (1) a hydrodynamic model





for flood simulation, (2) an agent-based model for evacuation management policies and
human behaviors, and (3) a transportation model for simulating household evacuation
processes in a road network. We apply the model to the Xiong'an New Area and examine
household evacuation outcomes under various shelter location plans and human behavior
scenarios. The results show that household evacuation processes are significantly affected
by the number and geographical distribution of evacuation shelters. Surprisingly, we find
that establishing more shelters may not improve evacuation results if the shelters are not
strategically located. We also find that low heterogeneity in evacuation preparation times
can result in heavy traffic congestion and long evacuation clearance times. If each
household selects their own shortest route without considering the effects of other evacuees'
route choices, traffic congestions will likely to occur, thereby reducing system-level
evacuation performance. These results demonstrate the unique functionality of our model
to support flood risk assessment and to advance our understanding of how the multiple
management and behavioral factors jointly affect evacuation performances.
**Keywords:**
Socio-hydrology; Flood management; Agent-based model; Emergency evacuation; Shelter
allocation

## 40    1. Introduction

Flooding is one of the most devastating natural disasters and can lead to significant
numbers of fatalities, social and economic disruptions, property and infrastructure damage,
and environmental degradation around the world (Smith and Matthews, 2015; McClymont



et al., 2020; Brunner et al., 2020; Tanoue et al., 2016; Kreibich et al., 2014; Wang et al.,
2019). The global flood database shows that the global flood inundation land area is
approximately 2.23 million km$^2$, with 255~290 million people being directly affected by
floods (Tellman et al., 2021). Flood-related economic damage increased globally from $94
billion in the 1980s to more than $1 trillion U.S. dollars in the 2010s (Hino and Nance,
2021). Furthermore, the severity, duration and frequency of damaging floods are expected
to continue to increase in the future due to changes in climate, land use and infrastructure
(Jongman et al., 2012; Moulds et al., 2021; Wedawatta and Ingirige, 2012; Tellman et al.,
2021). In many areas facing increasing flood threats, it is essential for emergency
responders and decision-makers to use advanced computer models to assess the flood risk
in flood-prone areas and to establish effective disaster-mitigation plans for informed flood
management (Simonovic and Ahmad, 2005).
Before an extreme flood occurs, evacuation is a critical emergency preparedness measure
and a common practice because it is impractical and/or economically costly to construct
the necessary infrastructure to resist floods (Wang et al., 2016; Liu and Lim, 2016; Islam
et al., 2020; Kreibich et al., 2015). However, studies have shown that emergency
evacuation is a complex and dynamic process that can be affected by factors from a wide
range of interdisciplinary domains (Zhuo and Han, 2020; Hasan et al., 2011; Huang et al.,
2012; Chen et al., 2021; Sung et al., 2018). These factors include, but are not limited to, (1)
the accuracy, lead time and sources of flood early warnings and the broadcasting channels
through which flood information is disseminated to the affected population (Shi et al., 2020;
Verkade and Werner, 2011; Alonso Vicario et al., 2020; Palen et al., 2010; Nester et al.,
2012; Goodarzi et al., 2019), (2) the infrastructure and engineering facilities needed for



emergency evacuation, which are influenced by the accessibility of transportation networks,
road capacity and location of evacuation zones (Mostafizi et al., 2017; Chen and Zhan,
2008; Saadi et al., 2018; Mostafizi et al., 2019; Koch et al., 2020; Oh et al., 2021; Liu and
Lim, 2016), and (3) demographical attributes and household behavioral characteristics,
such as residents' belief and risk perception, previous knowledge, social networks, and past
experience with flood events (Hofflinger et al., 2019; Huang et al., 2017; Lindell et al.,
2020; Wang and Jia, 2021; Shahabi and Wilson, 2014; Du et al., 2017). These studies
highlight the need to develop comprehensive socio-hydrological modeling tools that can
adequately incorporate various factors and processes to support flood management plans
in the context of coupled flood-human systems.
Among the many emergency management policies and plans, shelter location arrangement
is essential for massive evacuation operations. City planners and policy makers need to
identify safe areas outside of flood inundation region as feasible shelter locations for
households who live in the at-risk areas. There have been some studies that explored the
criteria of shelter location arrangement for evacuation planning (Alçada-Almeida et al.,
2009; Nappi and Souza, 2015; Bayram et al., 2015; Li et al., 2012; Alam et al., 2021). For
instance, Bayram et al. (2015) developed an optimization model to allocate evacuation sites
and assign each evacuee to the nearest shelter, with the objective of minimizing the total
evacuation time. However, in this study each evacuee's travel time is estimated based on a
simple traffic volume-travel time function, which is not able to fully represent evacuees'
complex interactions in a road network. Liu and Lim (2016) applied spatial analysis
methods to assign shelters to evacuating households, considering the spatial relationships
between households and shelter sites. A limitation of this study is that evacuee's travel time



is obtained from a simplified traffic model and the road network is not well represented in
the network analysis. In a recent study, Alam et al. (2021) used a massive traffic simulation
model and a multiple criteria evaluation method to identify candidate evacuation shelters,
taking into account of environmental conditions, structural attributes, emergency services
and transportation aspects. However, this study focused on obtaining a suitability score for
each candidate shelter site with various weighting factors, and failed to examine to what
extent evacuation performance could be affected by the number of shelters and their
geographical distribution in the community. Nevertheless, the current studies have left a
research gap that warrant research efforts to use physically-based flood simulation models
to identify safe areas as feasible shelter locations, and more importantly, to use
transportation models to systematically evaluate how evacuation performances could be
affected by the number and geographical distribution of evacuation shelter locations. This
is the primary research question we seek to explore in this study.
The second research question to be explored in this study is associated with the role played
by human behaviors in evacuation processes, which is an important research direction in
disaster management (Aerts et al., 2018; Simonovic and Ahmad, 2005; Urata and Pel,
2018). After receiving flood evacuation warnings, households will make decisions based
on flood risk information, spend some time to complete a set of preparation tasks, and then
evacuate from their homes to safe areas. Among these decisions and behaviors, households'
evacuation preparation times (i.e., from the time when they receive flood evacuation orders
to the time when they start to evacuate on road) play an important role in evacuation
performances. Many empirical studies have examined the geographic, demographical and
behavioral factors that affect households' preparation times (Lindell et al., 2005, 2020;





Huang et al., 2012, 2017; Chen et al., 2021). They found that household evacuation
preparation times could vary significantly from a household to another, exhibiting a certain
degree of behavioral heterogeneity in a community (Lindell et al., 2005, 2020; Rahman et
al., 2021). As a result, here we hypothesize that the heterogeneity in households'
evacuation preparation times affect the traffic flow on the road network and consequently,
affect the final evacuation outcomes. However, there are few studies that have explicitly
examined how traffic condition and evacuation performances are affected by different
degrees of heterogeneity in households' evacuation preparation times (Wang et al., 2016).
This is the second research question we aim to explore in this study.
Furthermore, in this study we also seek to examine how evacuation processes are affected
by households' evacuation route searching strategies, which is another question that
concerns emergency responders and policy makers. Previous studies have mostly applied
the shortest distance path searching method for evacuees to find evacuation routes from
their original locations to evacuation destinations (He et al., 2021; Bernardini et al., 2017;
Du et al., 2016; Li et al., 2022). However, each evacuee's searching for the shortest
evacuation path may not ensure system-level evacuation outcomes. In this study, we focus
on comparing the evacuation scenario in which each household chooses the shortest path
for evacuation with the scenario in which system-level global optimal routes are assigned
to the evacuees. Such comparative analyses are expected to provide policy implications in
terms of evacuees' route selections to improve evacuation performances during natural
disasters.
Motivated by the above research questions and knowledge gaps, in this study we develop
an integrated socio-hydrological modeling framework that couples (1) a physically-based



hydrodynamic model for flood inundation simulation, (2) an agent-based model (ABM)
for simulating flood management plans and human behaviors, and (3) a large-scale traffic
simulation model for simulating households' evacuation processes in a road network. We
apply the modeling framework to the Xiong'an New Area, a large residential area with a
high risk of flood in north China. Using a 100-year flood hazard as an example, a set of
scenario analyses are conducted to explore how residents' evacuation processes are jointly
affected by management policies (i.e., the number and geographical distribution of
evacuation shelter locations) and human behaviors (i.e., the heterogeneity in households'
evacuation preparation times and route searching strategies). This study aims to provide
both modeling and policy implications for researchers and emergency responders to
develop advanced socio-hydrological modeling tools for flood risk assessment and to
improve our understanding of how flood evacuation performances are jointly affected by
many management and behavioral factors.
The remainder of this paper is organized as follows. Section 2 presents the modeling
framework. Section 3 introduces the case study site, model construction and scenario
design. Section 4 presents the modeling results. Section 5 discusses the insights, limitations,
and future research directions of this study, followed by the conclusions in Section 6.
**2. Methodology**
This section introduces the integrated modeling framework of this study. As illustrated in
Figure 1, the modeling framework consists of three models: (1) an ABM for simulating
household decision-making and emergency responders' flood management policies, (2) a
transportation model for simulating residents' evacuation activities in a road network, and



(3) a hydrodynamic model for simulating flood inundation processes. Detailed introduction
to the three models and their coupling methods are described in turn as follows.

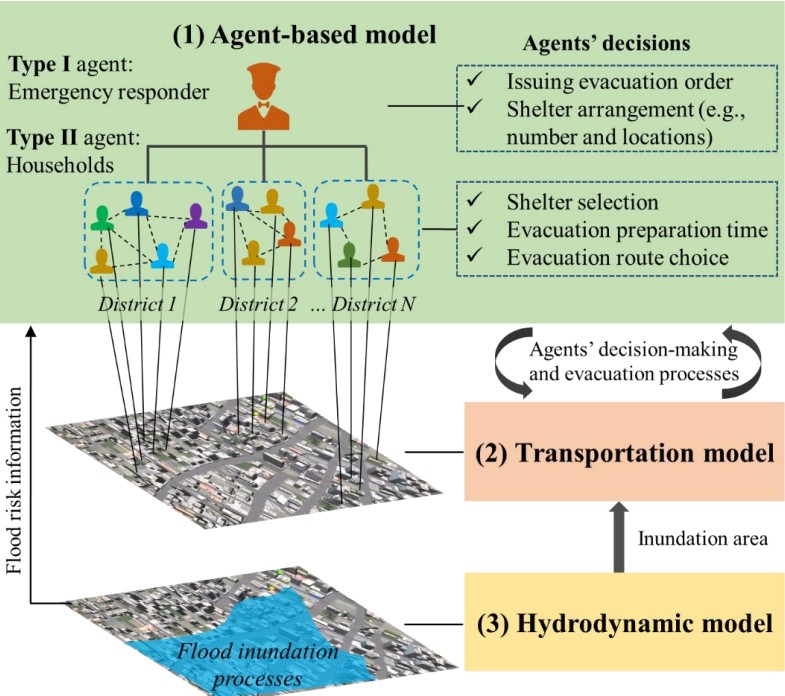


**Figure 1**. Illustration of the integrated modeling framework that couples an ABM for
simulating household decision-making and emergency responders' flood management
policies, a transportation model for simulating residents' evacuation processes in a road
network and a hydrodynamic model for simulating flood inundation processes
**2.1. The ABM for human decision-making during flood events**
In this study, an ABM is developed to simulate government's disaster management plans
and residents' flood evacuation behaviors. Therefore, two types of agents are considered
in the ABM: (1) an emergency responder (Type I agent) and (2) the set of households (Type
II agents), which are described in turn below.





The emergency responder agent represents a government institution that makes flood
management plans. As shown in Figure 1, in this study, we specifically consider two flood
management decisions: (1) issuing a flood evacuation order to the residents who live in
flood-prone area and (2) shelter arrangement (i.e., deciding the number and location of
evacuation zones that should be used to protect evacuees from flood hazards). Note that
other management practices (e.g., sandbagging and levee construction) are also important
flood management measures, which are not explicitly discussed in this study.
In this study, each household is represented by an autonomous decision unit (i.e., an agent),
considering that all the family members in a household typically evacuate in a shared
transportation mode after communicating with each other in arriving at a final evacuation
decision (Du et al., 2016). After receiving evacuation orders, an agent will spend some
time to complete a set of evacuation preparation tasks and then evacuate from its household
location to a chosen evacuation destination. The following three decisions and/or behaviors
are explicitly considered during this process.
The first decision is selecting an evacuation shelter if multiple optional shelters are
available. In this study, we assume that an agent will choose the evacuation destination
(i.e., shelter) that is located the shortest geographical distance from its residential location.
The second decision is associated with evacuation preparation activities (e.g., gather family
members, pack bags, board up windows, and shut off utilities). These activities are
aggregated and represented by a behavioral parameter called the evacuation preparation
time. This parameter measures how long it takes an agent to prepare for evacuation and is
indicated by the interval between the time when an agent receives an evacuation order and
the time when they start to evacuate via a road network. Previous studies have shown that



households' evacuation preparation times are influenced by both psychological and
logistical preparation tasks, which may vary among agents, with noticeable behavioral
heterogeneity even at the community scale (Lindell et al., 2020, 2005; Wang et al., 2016).
In this study, the heterogeneity in agents' evacuation preparation times is represented by
the variation (i.e., standard deviation) in all the households' evacuation preparation times,
and we explicitly examine the role of human behavioral heterogeneity in community
evacuation outcomes.
The third decision is related to agents' route selection strategy during evacuation processes.
In a complex road network, an agent may have multiple route choices from an origin to a
destination. In this study, we assume that each agent has good knowledge of the road
network in their community. Thus, two route search methods are incorporated into the
model as (1) the shortest distance route search method (Mode 1) and (2) the system
optimization-based route search method (Mode 2). In the shortest distance route search
method, each agent focuses on finding the shortest route from their current location to the
selected evacuation destination in the road network (Gallo and Pallottino, 1988; Fu et al.,
2006; Li et al., 2022). Notably, an agent seeks to reduce their evacuation time without
considering the effects of other agents' evacuation route selections. In comparison, the
optimization-based route search method adopts a heuristic iterative method to optimize
agents' collective evacuation routes so that system-level evacuation efficiency is achieved
(Zhu et al., 2018; He et al., 2021). Based on the above three decisions and behaviors, all
the agents' movements and interactions in the road network are incorporated into a
transportation model, which is described in the following section.





## 2.2. Transportation model for large-scale evacuation simulation


As mentioned in Section 2.1, after an agent decides to evacuate, it will move from its
household location to a chosen evacuation destination through the traffic network. During
evacuation processes, an agent interacts with other agents and with the environment to
adjust their movement in the road network over time. There are a number of modeling
platforms and software packages used to model agents' evacuation processes. These
include the Network Explorer for Traffic Analysis (NEXTA), the Transportation Analysis
and Simulation System (TRANSIMS), the Planung Transport Verkehr (PTV) VISSIM, the
City Traffic Simulator (CTS), and the Multi-Agent Transport Simulation model (MATSim)
(Mahmud and Town, 2016; Lee et al., 2014; Murray-Tuite and Wolshon, 2013).
This study applies MATSim to simulate agents' evacuation processes. MATSim is a widely
used open-source software for large-scale transportation simulation. The model can
provide detailed information about each evacuee's travel demand, traffic flow and
movement in a road network (Horni, 2016; Lämmel et al., 2009, 2010; Zhuge et al., 2021).
As shown in Figure 2, MATSim requires a variety of data as model inputs. The *plan* data
include the initial locations, evacuation destinations, and departure times of all agents, and
these data can be retrieved from agents' attributes and evacuation decisions in the ABM.
The *network* data describe the attributes of the road network, such as the geographical
structure of the road network, the number of lanes of each road, and road segment lengths
and speed limits. These data are available from local or regional government institutions
(e.g., the Department of Transportation) or from online data retrieval platforms such as
Open Street Map or Google Maps (Farkas et al., 2014). Finally, the *config* input includes
a model execution engine that defines a set of global model environments. Three modules,



namely, an execution module, a scoring module, and a replanning module, are incorporated
into MATSim for transportation simulation. This model has been widely used by
researchers and practitioners to support evacuation planning and simulation for various
types of natural disasters, such as earthquakes (Koch et al., 2020), hurricanes (Zhu et al.,
2018), tsunamis (Muhammad et al., 2021), and floods (Saadi et al., 2018). For more details
about MATSim and its applications in transportation simulation, see Lämmel et al. (2009)
and Horni (2016).

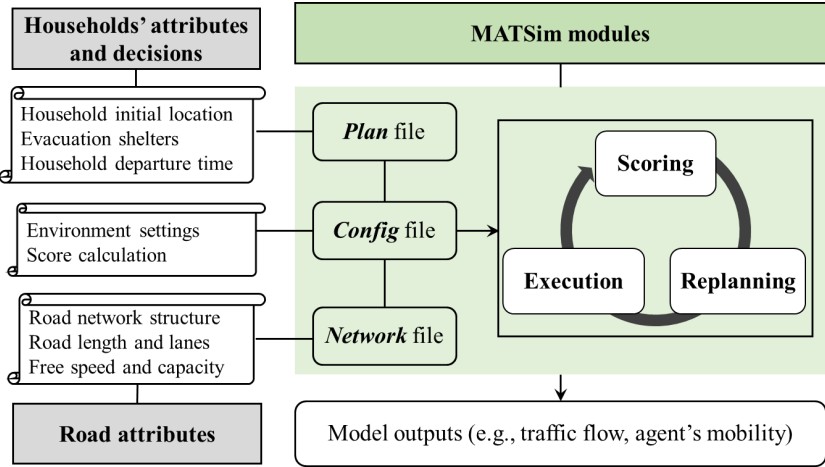


246             **Figure 2**. Input, modules and processes of the MATSim model

**2.3. The hydrodynamic model for flood inundation simulation**
Information on flood inundation processes (e.g., flood extent and water level) is essential
for governments and the public to make flood management and evacuation decisions.
Hydrodynamic models are important tools to simulate the timing and duration of flood
dynamics by solving a set of mathematical equations that describe fluid motion (Guo et al.,
2021). There are many hydrodynamic models available for flood dynamics simulation.





These include, but are not limited to, HEC-RAS, MIKE11, MIKE 21, JFLOW, TRENT,
TUFLOW and DELFT3D (Teng et al., 2017).
Following our prior work (Wu et al., 2021), in this study we use the classic hydrodynamic
model, MIKE 21, to simulate flood inundation processes in a floodplain. MIKE 21
numerically solves the two-dimensional shallow water equations to obtain water levels and
flows across space and over time in various watershed environments, such as rivers, lakes,
estuaries, bays and coastal areas. MIKE 21 has been widely used to simulate flood
inundation processes in many floodplains across the world, and is considered as one of the
most effective modeling tools for flood risk mapping, flood forecasting and scenario
analysis (Nigussie and Altunkaynak, 2019; Papaioannou et al., 2016). Interested readers
may refer to our prior work (Wu et al., 2021) for detailed introductions to the construction,
calibration and validation of MIKE 21 model in the specific study area.
**2.4. Model integration and flowchart of the modeling framework**
In the prior sections (Sections 2.1-2.3), the structures and functionalities of the three
models were introduced; this section introduces how they are coupled in an integrated
modeling framework. Previous studies have shown that computer models can be coupled
in either a loose or a tight manner (Harvey et al., 2019; Bhatt et al., 2014; Murray-Rust et
al., 2014; Du et al., 2020; Li et al., 2021). The former refers to models that are linked
together by input/output data interfaces. That is, the output of one model is used as the
input of another model. In contrast, for the latter, a model uses a common data pool and
workload to exchange data among multiple model components and, as a result, components
affect each other during model running processes.



In this study, both the loose and tight coupling methods are employed to combine the three
models. Specifically, MIKE 21 is coupled with the ABM and MATSim in a loose manner,
while the ABM and MATSim are coupled in a tight manner. The model coupling process
and flowchart of the integrated model are illustrated in Figure 3. First, MIKE 21 simulates
flood inundation processes for a specific flood event (e.g., a 100-year flood). The modeling
results of MIKE 21 are then used to assess the inundated area and affected households in
the flood zone, which are used as input data for the ABM and MATSim. Next, based on
the modeling results of MIKE 21, the two types of agents in the ABM are generated. The
household agents who are located in the flood zone will receive flood warnings from an
emergency responder agent and make evacuation decisions. Finally, all the agents'
movements and evacuation activities are simulated by MATSim. By integrating the three
models, the proposed modeling framework is capable of simulating flood inundation
processes, flood management practices, and household decision-making and evacuation
processes in a coherent manner. In the next sections, we will use a real-world case study to
demonstrate how the modeling framework can be used by researchers and practitioners for
flood risk assessment and evacuation management.



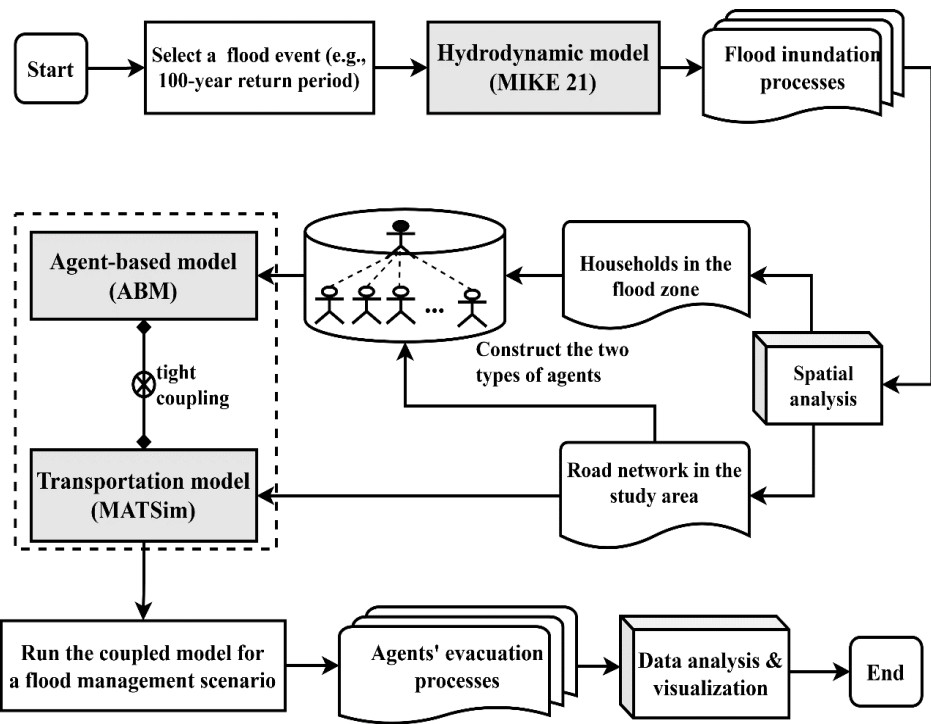


**Figure 3**. The flowchart of the integrated modeling framework

**2.5. Measurement of flood evacuation performance**
Agents' evacuation processes reflect their evacuation status and movements across space
and over time in a road network. In this study, we use multiple parameters and indicators
to represent agents' evacuation processes and evaluate their evacuation performance. For
a residential area with $n$ household agents, we first use a categorical variable, $S_{j,t} \in \{1, 2, 3\}$,
to describe an agent $j$'s evacuation status at time step $t$. $S_{j,t} = 1$ denotes that agent $j$ has not
started their evacuation process at time $t$. $S_{j,t} = 2$ denotes that agent $j$ has already started
evacuation but has not arrived at their evacuation destination at time $t$. $S_{j,t} = 3$ denotes that
agent $j$ has arrived at their evacuation destination at time $t$, which represents a successful

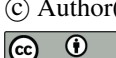



evacuation case. Let $\tau_0$ denote the time when the flood evacuation order is issued to the
public, and let $\tau_j$ and $\tau_j^*$ denote agent $j$'s departure time (i.e., the time when the agent starts
their evacuation in the road network after evacuation preparation time) and arrival time
(i.e., the time when agent $j$ arrives at their evacuation destination), respectively. The agent's
evacuation time $\phi_j$ is defined as the time period from their departure time $\tau_j$ to their arrival
time $\tau_j^*$ (i.e., $\phi_j = \tau_j^* - \tau_j$).
By summarizing all the agents' evacuation statuses over time, the effectiveness of flood
evacuation processes in a region can be reflected by a matrix with two indicators at the
system level: (1) agents' average evacuation time $\Phi$ and (2) the system-level evacuation
clearance time $\Gamma$. Agents' average evacuation time $\Phi$ is the average value of all the agents'
evacuation times, which is calculated by $\Phi = \dfrac{1}{n}\sum\limits_{j=1}^{n}\phi_j$. In comparison, the system-level
evacuation clearance time $\Gamma$ for a region is the duration from the time when the flood
evacuation warning is issued in the residential area to the time when the last agent arrives
at their evacuation destination (i.e., $\Gamma = \max(\{\tau_j^* \mid j = 1, 2, 3, ..., n\}) - \tau_0$).

**3. Case study and scenario design**

**3.1. Study site**

The Xiong'an New Area (XNA) is used as a case study to illustrate the functionality of the
proposed modeling framework in flood simulation and evacuation management. The XNA
is located in the Baiyangdian River Basin, which includes the largest freshwater wetland
in North China. This region covers three counties (i.e., Xiongxian, Rongcheng, and Anxin),

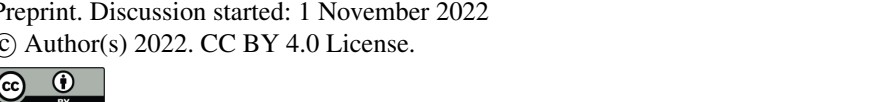

encompassing a total area of 1768 km$^2$ (Figure 4). The region has a population of 1.1
million, and the GDP is 21.5 billion RMB (Sun and Yang, 2019).
The XNA has a typical continental monsoon climate, with annual average precipitation
totaling approximately 570 mm. The region is influenced by various natural disasters and
environmental problems, such as water pollution, heat waves, and groundwater
overexploitation. In particular, the XNA has a high risk of flooding due to frequent extreme
rainstorm events (Jiang et al., 2018; Su et al., 2021). Historical climate records show that
a total of 139 flood events have occurred in the XNA over the past 300 years (Wang et al.,
2020). For example, the heavy storm from 19 July to 21 July in 2016 affected a total
population of approximately 517,000, leading to severe destruction and economic losses.
Studies have found that compared with historical flood conditions, both the frequency and
intensity of extreme flood events in the region are expected to increase under future climate
change (Zhu et al., 2017; Wang et al., 2020). The flood problems in the XNA and many
other flood-prone areas worldwide call for developing advanced computer models and
decision support systems for robust flood risk assessment and informed management
practices during extreme flood events.

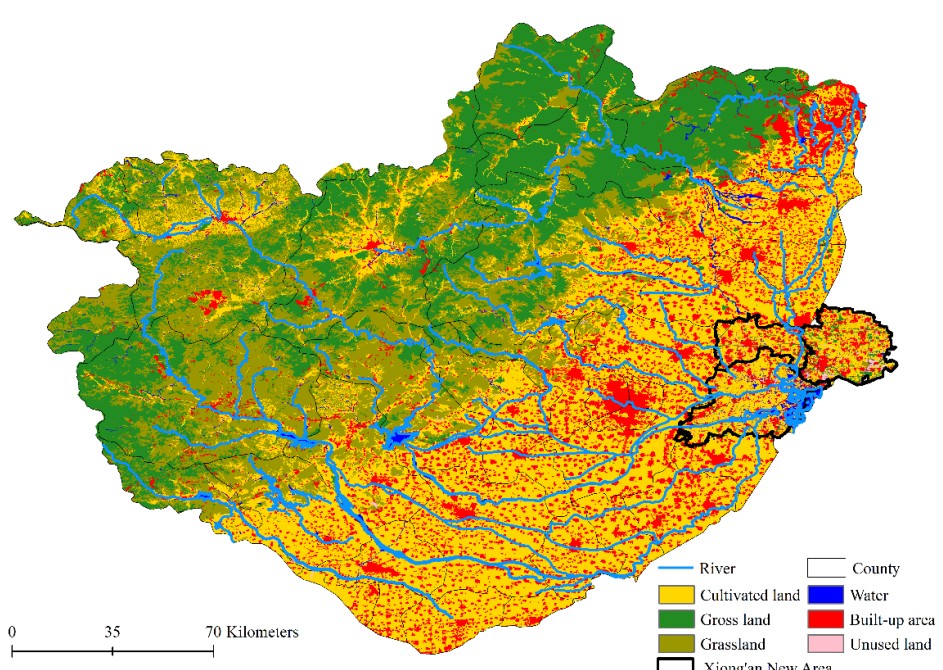

**Figure 4**. Map of the Baiyangdian River Basin and the Xiong'an New Area (marked with solid black lines)

## 3.2. Data collection and model construction

Based on the modeling framework, data from various sources were collected and compiled to construct the model, including meteorological, land-use, hydrological, transportation and census data. Among them, land topology is retrieved from the 7-meter resolution DEM from the State Bureau of Surveying and Mapping. Meteorological data (e.g., daily precipitation, temperature, solar radiation and wind speed) from 98 stations in the study area are collected from the China Meteorological Administration. Population and household distribution are based on 30-meter resolution census data from the census bureau





of local government. Road network data is retrieved from OpenStreetMap, an open source
global map data repository. Table 1 presents the primary data in this study and their sources.

Table 1. List of data used in the integrated model

| Data type | Data source | Period | Resolution | Format |
|---|---|---|---|---|
| Land elevation | State Bureau of Surveying and Mapping | 2019 | 7 m | TIF |
| Land use | Data Center of the Chinese Academy of Sciences | 2015 | 30 m | TIF |
| River network | Data Center of the Chinese Academy of Sciences | 2015 | - | SHP |
| Streamflow | Hydrological Yearbook in China | 1980-2010 | Daily | EXCEL |
| Weather conditions | China Meteorological Administration | 1980-2010 | Daily | EXCEL |
| Soil type | Data Center of Science in Cold and Arid Regions | 2009 | 1 km | TIF |
| Population | Census Bureau of the local government | 2020 | 30 m | EXCEL |
| Household distribution | Census Bureau of the local government | 2020 | 30 m | TIF |
| Road network | Open Street Map | 2022 | - | XML |


Figure 5 illustrates how the data are merged and integrated into the modeling framework.
As introduced in Section 2, the model starts by running the MIKE 21 model, with the
meteorological, DEM, land use, soil type and river network data as the model input. For a
given storm event, the MIKE 21 model generates flood dynamics processes, which can
predict the inundated area and the affected population. These data are then used to construct
the ABM and the MATSim model to simulate agents' flood management and evacuation
behaviors.



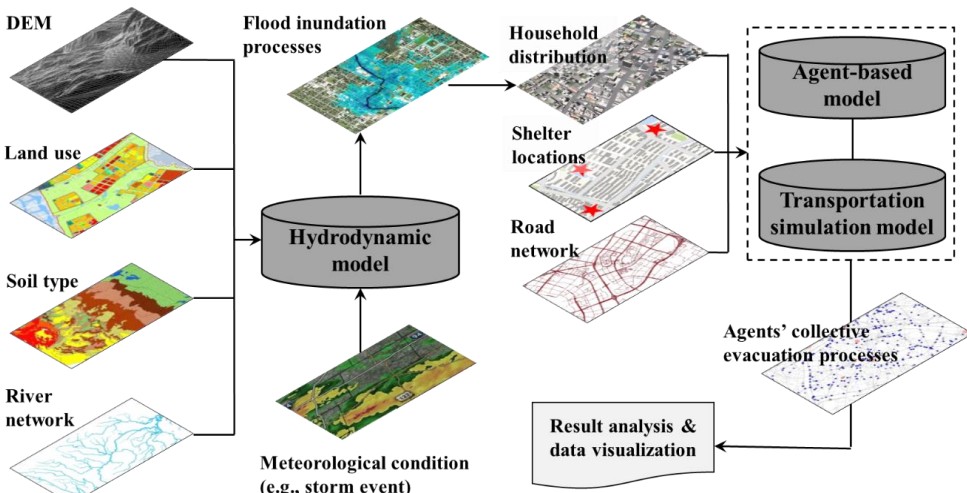


**Figure 5**. Data sources and flowchart of the integrated modeling framework
**3.3. Flood simulation and scenario design**
As mentioned above, the case study site has a high risk of flooding due to frequent extreme
rainstorm events. Following the precautionary principle in natural disaster management
(Etkin et al., 2012), we use the 100-year flood event as an example to evaluate the impacts
of extreme flooding on the study area, and then examine the role of various management
policies and human behaviors in household evacuation processes.
We run the hydrodynamic model to simulate flood inundation processes under the 100-
year return period. The modeling results show that the inundated area is 66.5% of the land
area in the 100-year return period (Figure 6). The affected population is 508,986 (45.8%
of the total population). These modeling results are consistent with the results that have
been reported in our prior work, and also agree with the empirical flood hazard experienced
by this region in July 2016. For detailed introductions to the construction, calibration and
validation of the hydrodynamic model, see Wu et al. (2021). With such a high flood risk,
it is essential for emergency responders to understand how flood evacuation performances
are affected by various human behavioral factors and evacuation management plans.

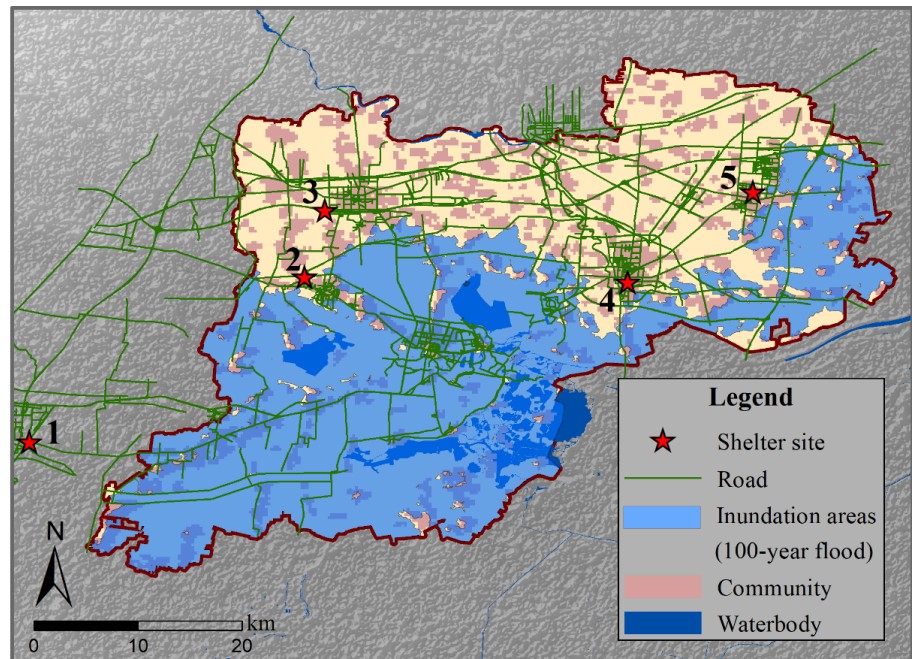


377          **Figure 6.** Flood inundation areas for the 100-year floods in the study area

A scenario-based analysis is conducted to examine the roles played by the following factors
in flood evacuation outcomes: (1) evacuation shelter establishment (i.e., the number and
geographical distribution of shelter locations), (2) heterogeneity in households' evacuation
preparation times, and (3) evacuees' route search strategies. Three experiments are
designed to assess the joint impacts of the above three factors (Table 2), which are
introduced in turn below.
The first experiment focuses on assessing the impact of the number and geographical
distribution of evacuation shelters on agents' evacuation processes. Note that in the XNA,



five optional sites for evacuation shelters are identified based on the flood inundation area
for the 100-year flood (illustrated by the red stars in Figure 6). Considering all the possible
combinations of these shelters, a total of 31 simulations are performed in this experiment
(i.e., 5 simulations for single-shelter scenarios and 26 simulations for multiple-shelter
scenarios). Experiment 2 assesses the impacts of agents' behavioral heterogeneity (i.e.,
variation in households' evacuation preparation times) on traffic flow and evacuation
outcomes. Note that in the first and second experiments, agents apply the shortest-distance
route search method (i.e., Mode 1) to evacuate from their household locations to evacuation
destinations. Experiment 3 simulates evacuation processes in which agents apply the
system-level optimization method (i.e., Mode 2) for route selection. The simulation results
of experiment 3 are compared with those of the first and second experiments to explore the
effects of agents' route search strategies on evacuation outcomes.

**Table 2.** Scenario design for simulating household evacuation processes

| Experiment | Shelter arrangement | Heterogeneity in agents' evacuation preparation times | Evacuation route searching strategy |
|---|---|---|---|
| 1 | All the combinations of the five optional shelters #1, #2, #3, #4, and #5 | 1.5[a] | Mode 1 (Shortest distance) |
| 2 | {#1, #2, #3, #4, #5}[b] | 0.2~3.0[a] | Mode 1 (Shortest distance) |
| 3 | Five one-shelter scenarios and {#1, #2, #3, #4, #5} | 0.2~3.0[a] | Mode 2 (System optimization) |

Note:

[a] Residents' behavioral heterogeneity is measured by the variation (i.e., standard deviation) in their evacuation preparation times. In the study area, residents' average evacuation preparation time is set to 4 hours based on our communication with the local flood management authorities.

[b] The set {#1, #2, #3, #4, #5} denotes that all five shelters are selected for this scenario.






## 4. Modeling results

### 4.1. An example of household evacuation processes

In this study, the results of household evacuation simulations are extracted and analyzed with a data visualization tool *Senozon Via* (Milevich et al., 2016). Figure 7a presents a snapshot of residents' evacuation schemes for the case in which all five evacuation shelters are used in the study area (note that each household is illustrated by a green dot in Figure 7a). Figure 7b depicts the change in the ratio of the three groups of the population during the evacuation processes. The percentage of the population in the S=1 group (i.e., the agents that have not started evacuating) displays a consistent decreasing trend, as more agents start their evacuation processes over time. Consequently, the S=3 group (i.e., the agents that have arrived in a safe zone) exhibits a consistent increasing trend. The S=2 group (i.e., the agents that have started evacuating but have not arrived at a safe zone, representing the residents who are moving in the road network) increases at the beginning of the evacuation period, reaching a peak of 43.1% after approximately 6.5 hours, and then decreases until the end of the evacuation period. The entire evacuation process takes approximately 15.5 hours (i.e., evacuation clearance time). In the following sections, the factors that influence the evacuation process will be assessed under different conditions.

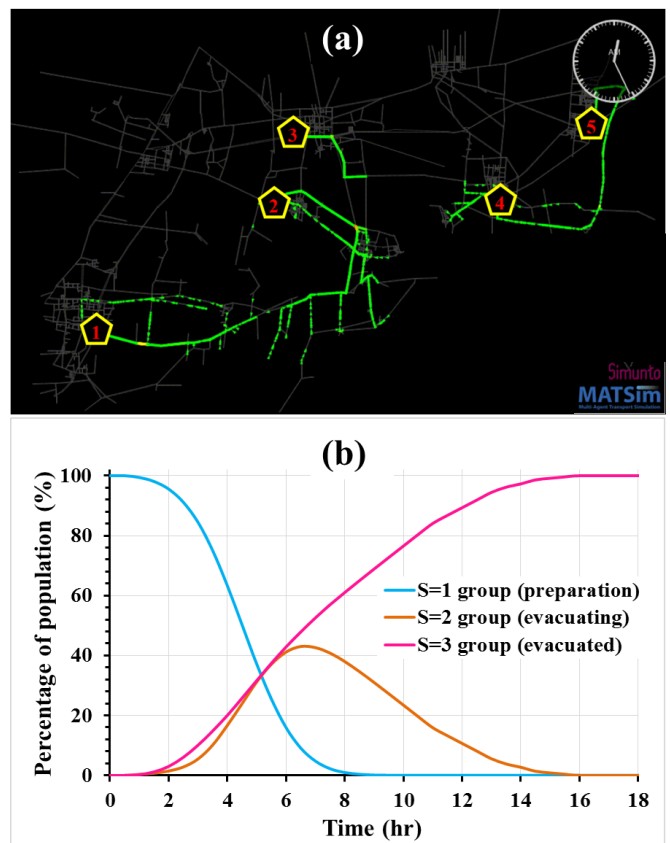

416

**Figure 7.** (a) A snapshot of residents' evacuation schemes when all five evacuation shelters

are established in the study area; (b) The percentages of the population in the three groups

of agents. Note that the S=1 group includes agents who have not started evacuating, S=2

includes agents who have started evacuating but have not arrived at an evacuation

destination, and S=3 includes agents who have successfully arrived at their destinations.

### 4.2. Impacts of shelter location arrangement on evacuation processes

We first conduct experiment 1 to examine agents' evacuation processes for the five

scenarios in which only one evacuation shelter is established. Figure 8 shows that the

geographical location of an evacuation shelter has a fundamentally important influence on



residents' flood evacuation performance. Residents' average evacuation time is the shortest
for shelter site #1 (20.1 hours), followed by sites #2 (23.7 hours), #5 (33.3 hours), #3 (35.7
hours) and #4 (46.8 hours). The boxplot of all the agents' evacuation times also shows that
the variation in agents' evacuation time is the largest for shelter site #4 (32.4 hours) and
the shortest for shelter site #1 (15.4 hours). In terms of the system-level evacuation
outcomes, shelter sites #1 and #2 are associated with the shortest evacuation clearance time
(~ 56 hours), and shelter site #4 is associated with the longest evacuation clearance time
(~108.9 hours) (the embedded figure in Figure 8). In this regard, among the five optional
shelter locations, sites #1 and #2 are the best locations for shelter establishment, and site
#4 is the worst, with the longest evacuation time.

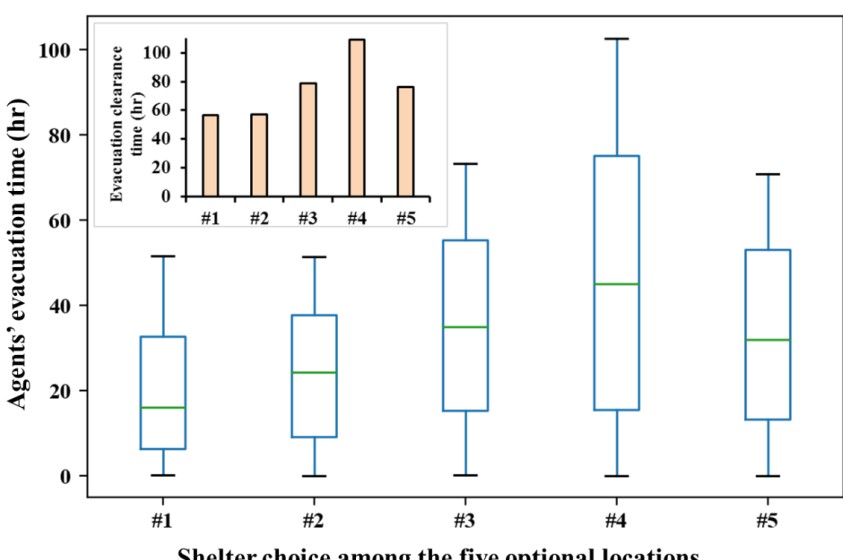


**Figure 8.** Boxplot of agents' evacuation times (the main figure) and the system-level
evacuation clearance times for the five one-shelter scenarios



Next, we compare the average evacuation time of agents for simulations in which all 31
combinations of the five optional evacuation shelter locations are considered. As shown in
Figure 9, when there are a small number of evacuation shelters, establishing more shelters
in the system can notably reduce agents' evacuation times, and this effect is more
noticeable for the worst shelter allocation scenario (illustrated by the blue line) than for the
best shelter allocation scenario (illustrated by the red line). For example, as the number of
shelters increases from two to three, the average evacuation time is reduced from 44.7
hours (shelter set {#4, #5}) to 29.7 hours (shelter set {#3, #4, #5}) for the worst shelter
allocation scenario (i.e., a total reduction of 15 hours). In contrast, the reduction in
evacuation time is only 5 hours for the best shelter allocation scenario (from 13.1 hours for
set {#2, #3} to 8.1 hours for set {#1, #2, #3}).

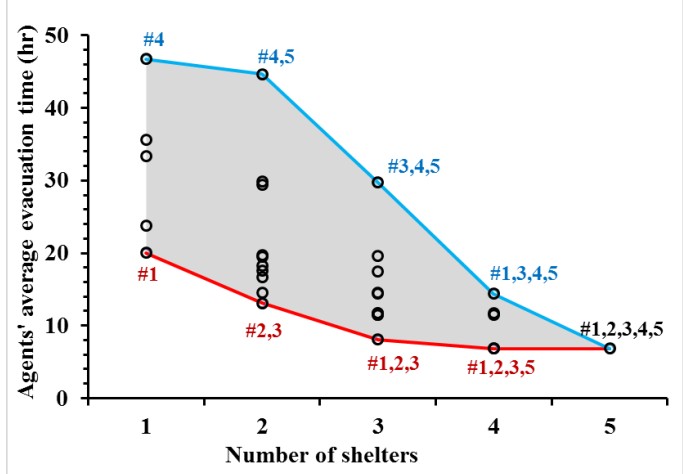


**Figure 9.** Residents' average evacuation time under the scenarios that consider all the
possible combinations of the five optional evacuation shelters

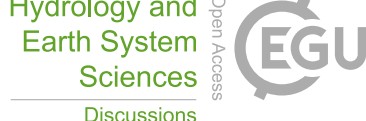

Notably, we find that the reduction in residents' evacuation time due to the increase in the
number of evacuation shelters is significantly affected by the existing number of
evacuation shelters and, in particular, their geographical distribution in the region. After a
certain number of evacuation shelters are established (larger than three in this case),
including more shelters in the system has a marginal effect in reducing evacuation times.
Taking the best shelter allocation scenario as an example (the red line in Figure 9), when
there are only two evacuation shelters ({#2, #3}), adding one more evacuation shelter (#1)
in the system can reduce the evacuation time by 5 hours (i.e., from 13.1 hours for set {#2,
#3} to 8.1 hours for set {#1, #2, #3}). In contrast, the reduction in evacuation time is only
1.3 hours when shelter #5 is added to the shelter set {#1, #2, #3}. In particular, it is noticed
that the average evacuation time is 6.8 hours for shelter sets {#1, #2, #3, #5} and {#1, #2,
#3, #4, #5}, which indicates that adding one more shelter in the system did not reduce the
average evacuation time. This phenomenon is supported by the Braess paradox phenomena
in the field of transportation research (Braess et al., 2005; Pas and Principio, 1997;
Murchland, 1970), which suggests that including a new link in a traffic network could
possibly result in heavier traffic congestion and longer travel times. This phenomenon and
its policy implications will be further discussed in Section 5.
**4.3. Impacts of residents' behavioral heterogeneity on evacuation processes**
Previous studies have shown that the evacuation preparation time of households plays an
important role in their emergency evacuation outcomes during natural disasters (Lindell et
al., 2005, 2020). However, the heterogeneity in human behaviors has not been explicitly
examined in flood evacuation processes. In this section, we conduct experiment 2 to assess
the impacts of human behavior heterogeneity (i.e., measured by the variance in agents'



evacuation preparation times) on evacuation processes. Figure 10 shows that human
behavioral heterogeneity has a nonlinear effect on agents' evacuation outcomes. Increasing
the heterogeneity in households' evacuation preparation times will result in reductions in
the average evacuation time and the system-level evacuation clearance time, and this effect
is more significant when the variation in the evacuation preparation time is small (< 1.5
hours). In particular, when the variation in preparation time is large (> 2 hours), the change
in the heterogeneity of preparation times will not notably affect the average evacuation
time or the system-level evacuation clearance time. These results are consistent with the
modeling results obtained from our prior work, which examined the role of heterogeneity
in residents' tolerance to flood risk during evacuation processes (Du et al., 2016).

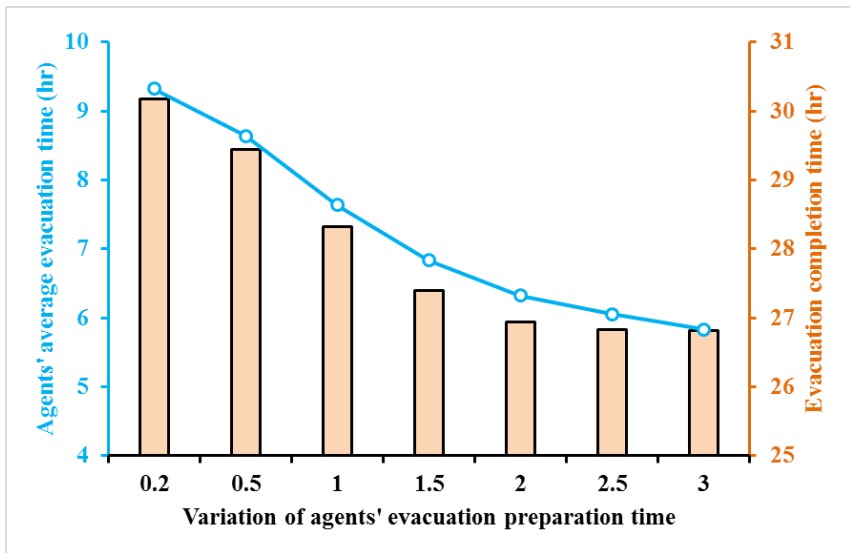


**Figure 10**. The impacts of human behavioral heterogeneity (i.e., the variation of agents'
evacuation preparation times) on their average evacuation time (the left Y-axis) and the
system-level evacuation clearance time (the right Y-axis)



Next, we assess the impacts of human behavioral heterogeneity on the traffic flow
conditions in the road network. Figure 11 plots the percentage of the three groups of the
population during evacuation processes, and the S=2 groups (illustrated by the two brown
lines) are the agents who are evacuating in the road network. The modeling results show
that the traffic peak time (i.e., the time when the number of agents in the road network
reaches a maximum during the evacuation period) is delayed as the level of agents'
behavioral heterogeneity increases. In addition, the percentage of agents in the road
network at the peak traffic time is significantly lower in the high behavioral heterogeneity
scenario than in other scenarios. For example, the traffic peak time can be delayed from
6.0 hours to 8.5 hours as the variation in the evacuation preparation times increases from
1.0 hours to 3.0 hours. At the time of the traffic peak, the percentage of agents in the road
network is reduced from 67.9% (the low-heterogeneity scenario) to 46.6% (the high-
heterogeneity scenario), and the system-level evacuation clearance time is reduced from
28.5 hours (the low-heterogeneity scenario) to 27 hours (the high-heterogeneity scenario).
Figure 12 compares the peak traffic time and the percentage of evacuating agents at the
peak time under various levels of heterogeneity in agents' evacuation preparation times.
The modeling results show that as agents' behavioral heterogeneity increases, flood
evacuation outcomes can be improved (i.e., the traffic congestion problem is alleviated, the
peak traffic time is delayed, and the evacuation clearance time is reduced).

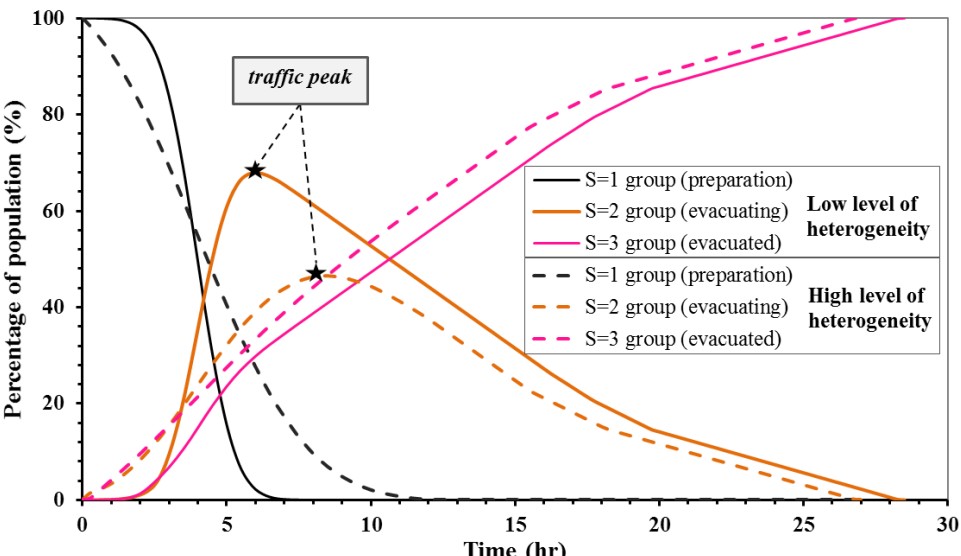

**Figure 11.** Comparison of the evacuation processes for low (solid lines) and high (dotted

lines) levels of human behavioral heterogeneity. Note that agent's behavioral heterogeneity

is measured by the standard deviation of their evacuation preparation time, and the low and

high levels of heterogeneity are 1.0 hours and 3.0 hours, respectively.

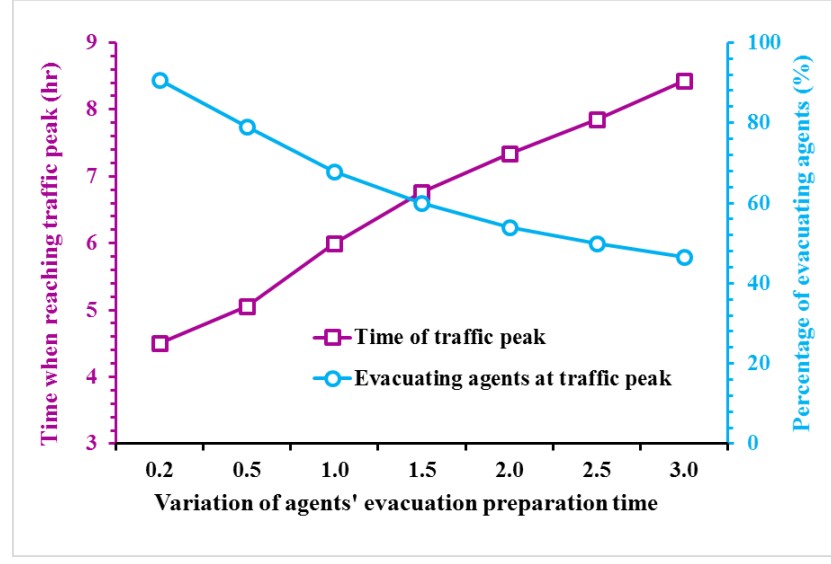





**Figure 12.** Peak traffic time (the left Y-axis) and the percentage of evacuating agents (i.e.,
S=2 group) at the peak traffic time (the right Y-axis) for various levels of human behavioral
heterogeneity.

### 4.4. Impacts of households' evacuation route choices on evacuation processes

In the above sections, the modeling results for scenarios in which the agents apply the
shortest-distance route search method to travel from their original locations to destinations
(i.e., Mode 1) during evacuation processes were presented. In this section, we conduct
experiment 3, in which agents' evacuation routes are obtained based on a system-level
optimization approach (i.e., Mode 2). Then, we compare the three experiments to explore
the joint impacts of the route search method and behavioral heterogeneity of residents on
evacuation processes.
Figure 13 compares agents' average evacuation times for the two travel modes. Two
implications are obtained from the modeling results. First, the results show that the average
evacuation time is consistently smaller for Mode 2 than for Mode 1. This result agrees with
the common belief in transportation research, in the sense that if each agent selects their
shortest evacuation route without considering the effects of other agents' route choices,
traffic congestion will likely occur in the road network. In contrast, if agents' evacuation
route choices are optimized from the system level, traffic flow conditions can be improved,
leading to a noticeable reduction in traffic congestion and shorter evacuation times.



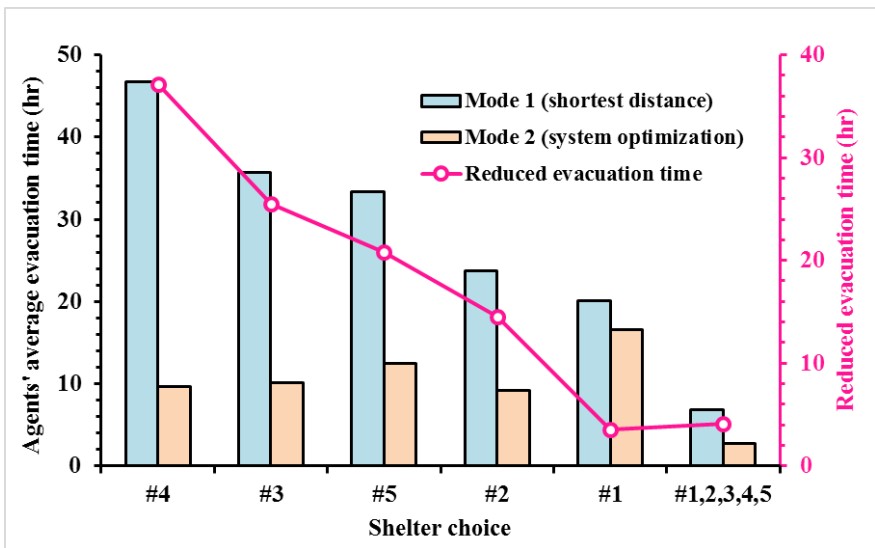


**Figure 13.** Comparison of the average evacuation time of agents for the two evacuation route search strategies

Second, one can observe that the variation in evacuation time across different shelter establishment scenarios is significantly higher for Mode 1 than for Mode 2. For example, among the five one-shelter scenarios, the agents' average evacuation time ranges from 46.7 hours to 20.1 hours (a difference of 26.6 hours) for Mode 1. In contrast, this value ranges from 16.5 hours to 9.2 hours (a difference of 7.3 hours) for Mode 2. This result implies that shelter establishment plays a more important role when residents only seek to minimize their individual evacuation times. In comparison, if agents' evacuation routes are optimized from the system level, shelter establishment will become a less significant factor affecting evacuation performance.

Figure 14 presents the percentages of the three groups of agents during the evacuation process, which aim to explicitly examine the impacts of different route search strategies. Compared with the shortest-distance search strategy (Mode 1), the system-level





optimization route search strategy (Mode 2) can reduce the evacuation clearance time by
12 hours (i.e., from 27.5 hours for Mode 1 to 15.5 hours for Mode 2). In addition, the
percentage of agents in the road network at the peak traffic time is reduced from 60.4% for
Mode 1 to 43.1% for Mode 2, indicative of a significant improvement in traffic congestion
during the evacuation period. However, the peak traffic time is similar in the two scenarios,
suggesting that changing agents' route search strategies does not considerably affect the
peak time of traffic flows.

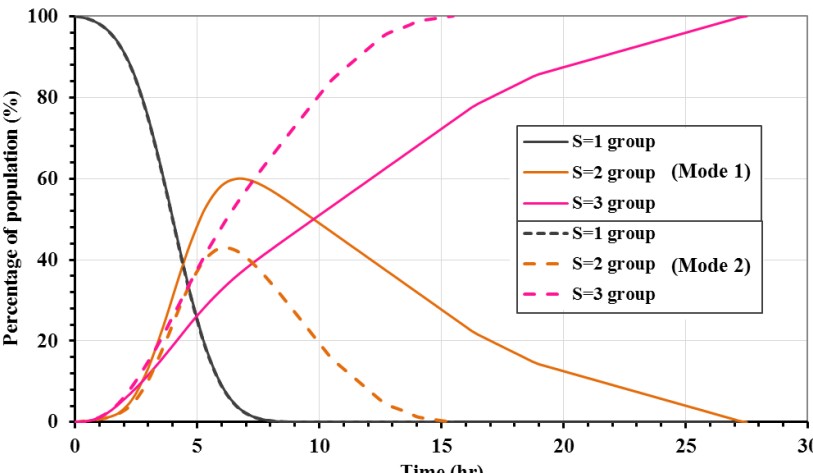


**Figure 14.** Comparison of residents' evacuation processes for the two route search
strategies (note that all five evacuation shelters are selected for the two scenarios, and the
variation in residents' evacuation preparation times is 1.5 hours)
The above analyses focused on assessing the impacts of a single factor (i.e., agents'
behavioral heterogeneity or evacuate route search strategies). Figure 15 examines how the
two factors jointly affect evacuation processes. Notably, in general, the average evacuation
time of agents and the system-level evacuation clearance time are small when the variation





in the evacuation preparation time is low and/or when agents follow Mode 2 to determine
their evacuation routes. Interestingly, when the variation in agents' evacuation preparation
times is low (<1.0 hour), the difference between Mode 1 and Mode 2 is not significant in
terms of the peak traffic time or the percentage of evacuating agents at the peak traffic time.
This result indicates that changing agents' route search strategies will not considerably
affect the peak traffic time or the maximum traffic flow if all the agents start their
evacuation activities within a short time window. In contrast, as the variation in the
evacuation preparation time of agents increases, the evacuation route search strategy used
can significantly affect the peak traffic time and the maximum traffic flow (Figures 15c-
15d). However, the variation in agents' evacuation preparation times does not notably
affect the changes in the average evacuation time or system-level evacuation clearance time
between the two route search strategies.

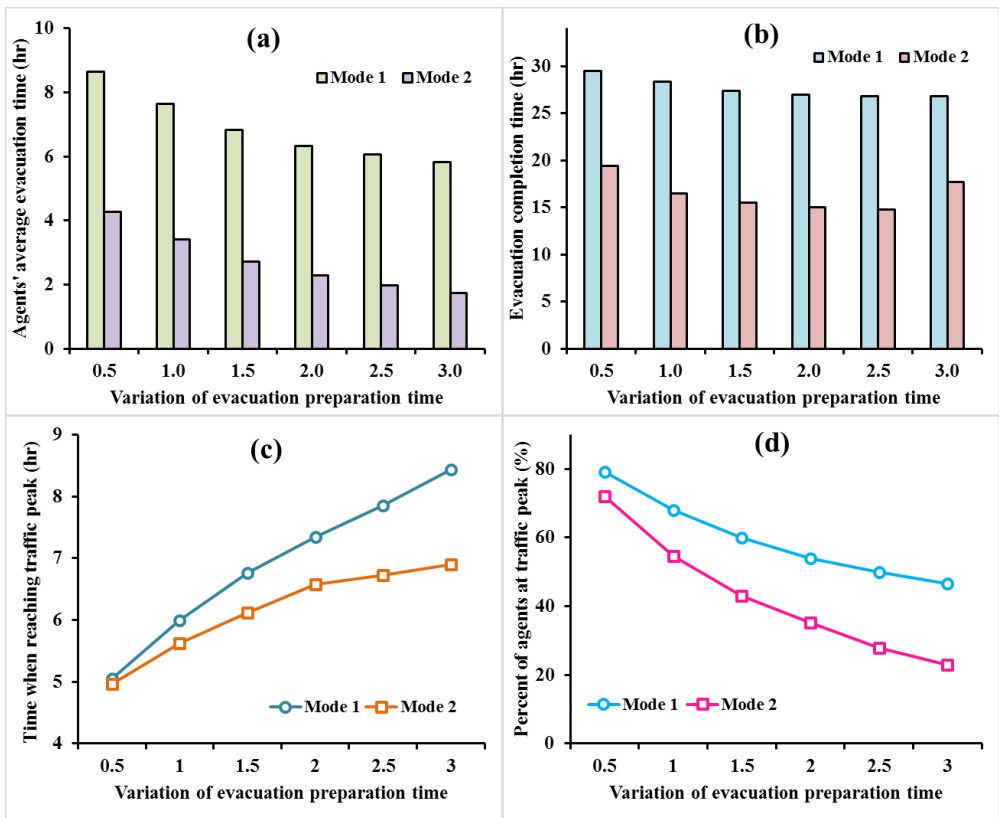

**Figure 15.** The joint impacts of evacuation route search strategies and the variations in agents' evacuation preparation times on (a) the average evacuation time, (b) the system-level evacuation clearance time, (c) the time when the traffic peak is reached during evacuation processes, and (d) the percentage of evacuating agents at the peak traffic time

## 5. Discussion

### 5.1. Implications for flood risk assessment and evacuation management

In this study, we employ an interdisciplinary socio-hydrological approach that incorporates a physically based hydrodynamic model, an agent-based human behavior and decision-making model, and a large-scale transportation model into an integrated modeling





framework. We apply the model to the Xiong'an New Area (XNA) in China to assess the
inundated areas of an extreme flood event and to examine household evacuation outcomes
under various management policies and human behaviors. Several modeling and policy
implications can be obtained based on the model construction and simulation results.
First, the simulation results of this study show that the flood risk of and the flood damage
to an area are not only affected by the hydrological characteristics of flood events but also
by infrastructural, socioeconomic and human behavioral factors. In particular, the results
show that household evacuation outcomes are significantly affected by shelter location
arrangement, route selection strategies, and evacuation preparation times. Therefore, it is
essential for researchers and policy makers to incorporate various social, hydrological and
human behavioral factors into an integrated framework to obtain more robust estimations
of flood risk and to design informed policies to support holistic flood management.
Second, the modeling results show that the number of evacuation shelters and, in particular,
their geographical distributions have important effects on flood evacuation processes. For
example, by comparing the evacuation outcomes obtained for the five optional shelter sites
in the case study area, we find that the average evacuation time of residents varies from
20.1 hours (shelter site #1) to 46.8 hours (shelter site #4) (Figure 8). In this regard, if there
are limited available resources and only one evacuation site can be established in the area,
shelter #1 would be a better site than shelter #4 if the management goal is to minimize the
average evacuation time of residents. Another implication associated with shelter choice is
that establishing more shelters in the area does not necessarily lead to improvements in a
community's evacuation processes if there is already a sufficient number of evacuation
shelters or if the shelters are not well distributed in the region. For example, in the case in





which there are three shelters (e.g., {#1, #2, #3}), including more shelters in the system
(e.g., #4, #5, or both) will not effectively reduce households' the average evacuation time
(Figure 8). This finding, although somewhat contrary to what one would intuitively expect,
is in line with the classic Braess paradox in the field of transportation research; notably,
adding a new link in a traffic network may not improve the operation of the traffic system
(Frank, 1981; Murchland, 1970). Some studies have shown that the occurrence of Braess
paradox phenomena may be affected by the road network configuration, travel demand,
and travelers' route search behaviors (Pas and Principio, 1997; Braess et al., 2005).
Therefore, regarding emergency management policies such as where to establish more
shelters, policy-makers need to scrutinize the relationships among these factors to
determine the number and geographical distributions of shelters in the system.
Third, flood evacuation is a complex process in which residents' evacuation activities can
be affected by various social, economic, environmental and infrastructural factors. Thus,
in a particular flood-prone area, residents' decisions and evacuation behaviors could be
highly heterogeneous, varying from family to family, from community to community, and
from time to time (Paul, 2012; Huang et al., 2017). This study shows that human behavioral
heterogeneity can significantly affect the flood evacuation outcomes in a given region. For
example, the modeling results show that variations in residents' evacuation preparation
times could result in noticeable differences in traffic congestion conditions and the time
required for evacuees to complete their evacuation processes (Figures 10-12). Therefore,
in flood management practice, emergency responders need to explicitly consider the
heterogeneity in residents' behaviors and determine how to promote behavioral changes
by providing the needed resources to vulnerable groups who are not able to take effective





flood mitigation actions to improve the overall disaster management performance in the
community (Nakanishi et al., 2019; Hino and Nance, 2021).

### 5.2. Limitations and future research directions

Our modeling framework and the simulations in this study have a number of limitations
that warrant future research to make improvements and extend the current approach. First,
similar to other studies on emergency evacuation simulation (Wood et al., 2020; Zhu et al.,
2018; Koch et al., 2020; Saadi et al., 2018), this study focuses on car-based traffic
simulation without considering other transportation modes (e.g., motorcycles). In real-
world evacuation cases, residents may use various types of transportation modes to
evacuate, including by automobile, motorcycle, bus, or on foot (Melnikov et al., 2016).
Residents may also change their travel modes during evacuation processes, for example,
due to a change in the available transportation facilities. Recent studies have attempted to
improve emergency evacuation simulations by considering more factors in evacuation
simulation, such as multiple transportation facilities, changes in traffic network
accessibility, variations in travel demand, pedestrian/vehicle interactions and speed
adjustments (Dias et al., 2021; Takabatake et al., 2020; Wang and Jia, 2021; Sun et al.,
2020; Chen et al., 2022). Future research can extend upon this study by incorporating these
factors into the modeling framework.
Second, regarding the analyses of shelter establishment, we primarily focus on the number
and geographical distribution of evacuation shelters without considering other important
shelter characteristics, such as shelter capacity. However, it is sometimes necessary to
consider the constraint of shelter capacity in evacuation management, especially in large-
scale evacuation scenarios. Recently, studies have analyzed the impacts of shelter





capacities and their geographical distribution on evacuation outcomes (Alam et al., 2021;
Khalilpourazari and Pasandideh, 2021; Oh et al., 2021; Liu and Lim, 2016). Future studies
should consider more shelter properties to improve the current modeling framework.
Third, in this study, the hydrodynamic model is coupled with the agent-based model and
transportation model in a one-way coupling manner. That is, the hydrodynamic model
generates flood inundation results as the input for the agent-based model and transportation
model, but the modeling results of the agent-based model and transportation model do not
affect the hydrodynamic modeling processes. Such a one-way model coupling method is
suitable for simulating residents' evacuation activities before a flood occurs, but it is not
suitable for cases in which evacuation processes and flood inundation processes have an
overlapping time period. In particular, the model is not capable of simulating how human
behaviors affect flood inundation processes, which is another limitation that needs to be
addressed in future work.

## 6. Conclusions

A fundamental aspect of societal security is natural disaster management. Computational
models are needed to assess the flood risk in flood-prone areas and to design holistic
management policies for flood warning and damage mitigation. In this study, we propose
an integrated socio-hydrological modeling framework that couples a hydrodynamic model
for simulating flood inundation processes, an agent-based model for simulating the flood
management practices of emergency responders and human behaviors, and a large-scale
transportation model for simulating household evacuation processes in a road network.
Using a case study of the Xiong'an New Area in China, we demonstrate the effectiveness
of the modeling framework for assessing flood inundation processes for a 100-year flood





event and examining households' evacuation outcomes considering various evacuation
management policies and human behaviors. A number of scenario analyses are performed
to explore the impacts of shelter location arrangement, evacuation preparation times and
route search strategies on evacuation performance.
Through a set of scenario analyses, the modeling results show that for a 100-year flood
event, approximately 66.5% of the land area will be flooded, affecting 0.5 million people.
Household evacuation processes can be significantly affected by the number and
geographical distribution of evacuation shelters. For the five optional sites of evacuation
shelters, the average evacuation time of residents ranges from 20.1 hours to 46.8 hours,
depending on where the evacuation shelter is located. Counterintuitively, yet in line with
the Braess paradox in the transportation field, we find that including more shelters in the
system may not improve evacuation performance in a region if the number of shelters or
shelter distribution is already optimal or near optimal. In addition, the simulation results
show that residents' flood evacuation outcomes are significantly affected by human
decision-making processes, such as the selection of evacuation route search strategies.
Compared with the system-level route optimization method, the shortest-distance route
search method is associated with a longer evacuation travel time because evacuees seeking
to minimize their own travel time may experience traffic congestion. We also find that a
low level of heterogeneity in agents' evacuation preparation times can result in heavy
traffic congestion and long evacuation clearance time. These modeling results highlight
that the flood risk of, and the ultimate damage to, an area is affected not only by the level
of the flood itself but also by flood management practices and household behavioral factors.
This study is therefore in line with some previous studies that highlight the significance of



a socio-hydrological approach for water science and watershed management (Di
Baldassarre et al., 2013; Sivapalan et al., 2012; Abebe et al., 2019).
This study still has a number of limitations that need to be addressed. Recommended future
work includes incorporating more types of transportation facilities into the transportation
model, considering the role of shelter capacity in evacuation management, and improving
the model coupling method by employing a two-way coupling approach to simulate the
impacts of human behaviors on flood inundation processes. We envision that these
extensions will improve the functionality of the proposed modeling framework, and the
simulation results with these improvements can provide more useful modeling and policy
implications to support flood risk assessment and emergency evacuation management.

**Acknowledgments**
Financial support from the National Natural Science Foundation of China (grant numbers
41971233, 51909118, and 41861124003) is gratefully acknowledged.

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
