# Peer review of "Development of an integrated socio-hydrological modeling framework"

_Hydrology and Earth System Sciences, 2022_

## Author Comment (AC1)

**Response to Reviewer #1's comments on the manuscript HESS-2022-362**

**General comments:**

This paper develops an integrated socio-hydrological modeling framework that couples a hydrodynamic model, an agent-based model, and a transportation model to examine household evacuation outcomes under various shelter location plans and human behavior scenarios. The results demonstrate the unique functionality of the model to support flood risk assessment and to advance the understandings of evacuation performances. The manuscript is well organized and written. The logic flow is easy to follow. Tables and Figures are clear and well presented. I think this is a high-quality manuscript, which will contribute to the flood management practice. I have only few minor concerns as follows:

**Response:**

Thank you very much for the positive comments and excellent feedbacks that have led to significant improvements to this work. We have addressed your comments point-by-point as follows. Note that the text in grey are the reviewers' original comments. The text in blue are our responses to the comments. The text in red are the new additions included in the revised manuscript.

**Comment 1:**

Lines 184-187: May agents also consider the shelter with least travelling time? please check the assumption.

**Response to comment 1:**

Thank you for the comment, which helps us to clarify the assumptions of this study. Yes, the agents will seek to evacuate to the safe areas as soon as possible, aiming to minimize total traveling time during evacuation processes. However, during an emergency situation, it is unclear and/or quite challenging for the agents to assess which shelter can ensure the shortest traveling time due to, for example, uncertainties of real-time traffic condition and traffic load (e.g., the number of evacuating agents on the road). Therefore, we follow the classic approach in evacuation simulation and assume that an

agent focuses on choosing the shortest route from its original location to the safe area, thereby choosing a closer shelter in the system as its evacuation destination (Note that a shorter traveling distance is typically associated with a shorter traveling time). In the revised manuscript, we have followed the comment and added some text to elaborate the assumption, which read as follows.

"During flood evacuation processes, the agents seek to evacuate to safe areas as soon as possible, aiming to minimize their traveling times. However, during an emergency situation, it is unclear and/or quite challenging for the agents to assess which shelter can ensure the shortest traveling time due to, for example, uncertainties of real-time traffic condition and traffic load (e.g., the number of evacuating agents on the road). Therefore, we follow the classic approach in evacuation simulation and assume that an agent focuses on choosing the shortest route from its original location to the safe area, thereby choosing the geographically nearest shelter in the system as its evacuation destination." (Lines 194-202)

**Comment 2:**

Lines 209-212: Will family agents consider at system level? Why will agents want to contribute to system efficiency? Mode 2 should be re-interpreted based on rational assumption.

**Response to comment 2:**

Thank you for the question that helps us to explain and clarify the motivation of analyzing the two route search modes. Yes, the agents will typically focus on reducing their own traveling times, and do not necessarily consider system efficiency during evacuation processes. Thus, mode 1 represents the case in which every agent focuses on its own evacuation efficiency (i.e., chooses the shortest route for evacuation), while mode 2 represents the case of system-level evacuation efficiency (i.e., all the agents' route choices are optimized at the system level). In this regard, mode 1 is the baseline evacuation scenario and mode 2 is the benchmark scenario. The results of mode 2 can be used to assess the extent to which the evacuation outcomes of model 1 can be

improved by changing agents' route choices. Policy makers can compare the results of the two traveling modes and then improve flood evacuation outcomes by, for example, providing route recommendations for the agents who may encounter/cause severe traffic congestion during their evacuation processes. We have followed the comment and included the following text in the revised manuscript to elaborate the motivation of analyzing the two travel modes.

"It is worth noting that the agents will typically focus on reducing their own traveling times, and do not necessarily consider system efficiency during evacuation processes. Among the above two route search modes, mode 1 represents the case in which every agent focuses on its own evacuation efficiency (i.e., chooses the shortest route for evacuation), while mode 2 represents the case of system-level evacuation efficiency (i.e., all the agents' route choices are optimized at the system level). In this regard, mode 1 is the baseline evacuation scenario and mode 2 is the benchmark scenario. The results of mode 2 can be used to assess the extent to which the evacuation outcomes of model 1 can be improved by changing agents' route choices. Policy makers can compare the results of the two traveling modes to improve flood evacuation management by, for example, providing route recommendations for the agents who may encounter and/or cause severe traffic congestion during evacuation processes." (Lines 235-246)

**Comment 3:**

In the results section, I think it is better to discuss the specific policy implications and recommendations following each result, from the perspectives of both emergency responders and family agents. In this way, readers can easily link the new findings to management practice.

**Response to comment 3:**

Thank you for the excellent suggestion. In the original manuscript, modeling results and discussions on policy implications are separated and presented in two sections (Section 4 for results and Section 5 for discussions). We agree with the reviewer that readers may be confused about how the results and policy implications are connected.

In the revised manuscript, we have followed the suggestion and included additional text in the result section to explicitly discuss their policy implications. The new additions are as follows.

"These results can yield policy implications in terms of the number and geographical locations of evacuation shelters needed to meet a particular flood management goal. For example, if the management goal is to evacuate all the residents to a single site, shelter #1 would be the best choice, among the five optional locations, in terms of minimizing the evacuation clearance time. However, for the case of establishing two shelters in the region, shelter set {#2, #3} is a better choice as compared with the other shelter site combinations." (Lines 483-489)

"These modeling results highlight the importance for policy makers to pay explicit attention to households' behavioral heterogeneity during flood evacuation processes. For example, the modeling results show that the variation in agents' departure times can significantly affect traffic load in the road network and evacuation clearance time. Traffic congestion condition can be alleviated if the variation of agents' departure times is larger. Thus, to improve evacuation efficiency, emergency responders may need to divide all the households in the community into a number of groups and guide them to evacuate in batches, rather than let them start evacuation in a chaotic manner without appropriate coordination." (Lines 552-560)

"The comparisons of the two route search methods show that households' route choices play an important role in their evacuation processes. Evacuation clearance time and traffic congestion will be significantly alleviated and become more robust against the change in shelter location arrangement if evacuation routes are optimized. In this regard, policy makers may improve flood management by providing clear guidance to all the households in terms where (i.e., shelter choice), when (i.e., departure time) and through which route (i.e., route selection) to evacuate in emergency conditions. In order to improve evacuation efficiency, households need to follow the evacuation guidance and take the recommended routes to travel to safe areas." (Lines 628-636)

---

## Author Comment (AC2)

**Response to Reviewer #2's comments on the manuscript HESS-2022-362**

**General comments:**

Summary: the authors evaluate tradeoffs between behavioral heterogeneity in departure time, which route they choose and where evacuation centers are location on traffic congestion and time to evacuate the affected population using a coupled model of flood inundation and traffic routing. The paper is very interesting and well organized. Following are few of my comments that the authors may want to consider before resubmitting their manuscript:

**Response:**

Thank you for the positive comments and excellent suggestions that have led to significant improvements to this study. We have addressed your comments point-by-point as follows. Note that the text in grey are the reviewers' original comments. The text in blue are our responses to the comments. The text in red are the new additions included in the revised manuscript.

**Comment 1:**

- at places authors may want to check for language

**Response to comment 1:**

We have followed the suggestion and have the paper proofread by a professional language editor from the *AJE Company*, which has very good reputation in English language editing for technical and academic writing. The changes have been included in the revised manuscript.

**Comment 2:**

- there is reference to sociohydrology - perhaps the authors can spend some space on why it is sociohydrology (e.g. because of bidirectional feedbacks between agents decisions and travel times). Also more references, placing this study in the landscape of other sociohydrological studies would be helpful.

**Response to comment 2:**

Thank you for the helpful suggestion. In the revised manuscript, we have added some text to describe why this model is a socio-hydrological model and how it is connected with other socio-hydrological studies, which read as follows.

"Specifically, the hydrological component of the socio-hydrological modeling framework is represented by the MIKE 21 model, which simulates flood inundation processes across space and over time in a flood-prone area for a given storm event. The simulation results of the MIKE 21 model can provide flood risk information and will be used by policy makers to make flood management plans. In comparison, the social component of the modeling framework is represented by the ABM and MATSim models, which simulate policy makers' flood management plans, households' responses to flood information and management plans, and their collective evacuation activities in the road network. By coupling the three models, our modeling framework is capable of simulating a wide range of components and processes in a coherent manner to support flood management." (Lines 138-147)

"The proposed modeling framework is motivated by and based on previous socio-hydrological studies that called for incorporating multiple factors in coupled human-flood systems to support holistic flood management. These factors may be associated with a wide range of interdisciplinary domains, such as hydrogeological conditions, flood inundation process, information dissemination platforms, risk perception and awareness, social preparedness, public policy, and urban infrastructure development (Barendrecht et al., 2019; Di Baldassarre et al., 2013; Fuchs et al., 2017; Pande and Sivapalan, 2017; Troy et al., 2015; Viglione et al., 2014; Yu et al., 2022)." (Lines 646-654)

**Comment 3:**

- It remains a semi-empirical study. The authors may want to discuss what next steps should be taken to make it more realistic in terms of mapping household behavior. For example, using household surveys on psychological factors that may influence such behavior. Can the behavior of others influence the psychology of those who have not

yet started to evacuate (e.g. "others are evacuating with urgency so I better hurry"). This may be a more conscious feedback than the travel time congestion feedback due to heterogeneity in time of departure)

**Response to comment 3:**

Thank you for the helpful comments. We fully agree with the reviewer that it would be helpful to consider more psychological and social factors for the improvement of this study. In the revised manuscript, we have added some text to discuss a number of future research directions, which read as follows.

"It is worth noting that this study is subject to many simplifications and assumptions due to data incompleteness and the specific research scope of the current work. Future study could incorporate more psychological and social factors to describe agents' decisions during evacuation processes. For example, future study can conduct surveys and questionnaires to quantify households' evacuation preparation times after receiving flood evacuation orders (Lindell et al., 2020). Also, future studies could consider other factors that may affect human flood risk perception and risk awareness, such as social memories, social interactions and observations of neighbors' actions (Du et al., 2017; Girons Lopez et al., 2017). These extensions and improvements can make the model capable of simulating more realistic decision-making processes and more complex human-flood interactions to support emergency management during floods." (Lines 745-755)

**Comment 4:**

- Table 1 is secondary not primary data. Primary data is self-measured, e.g. through field campaigns

**Response to comment 4:**

Thank you for clarifying the terminology. We have followed the comment and removed the word "primary" in the revised manuscript. (Line 383)

**Comment 5:**

- How are the travel time results affected when shelter locations are designed to be located close to denser parts of the population than when they are randomly assigned in space? Here, perhaps simulations with more number of shelters and where they are designed to be located are needed to conclude that marginal gains reduce as number of shelters are increased.

**Response to comment 5:**

Thank you for the comments about the relationship between agents' travel time and the number of shelters. Yes, the simulation results have shown that agents' travel time decreases if shelters are located closer to denser residential areas, because such a shelter location distribution method can reduce agents' travel distances as compared with the scenario in which shelters are randomly located. Indeed, the reviewer is correct that marginal gains of establishing more shelters will decrease as the number of shelters increases (shown by Figure 9). In the revised manuscript, we have followed the comments and added some text to highlight these findings, which read as follows.

"Notably, the modeling results show that agents' evacuation time decreases if shelters are located closer to denser residential areas. This is because a shelter located close to denser areas can reduce agents' travel distances (A shorter distance is typically associated with a shorter travel time). Furthermore, the modeling results show that the reduction in residents' evacuation times, due to the increase in the number of evacuation shelters, could be significantly affected by the existing number of evacuation shelters and, in particular, their geographical distribution. After a certain number of evacuation shelters are established (larger than three in this case study), including more shelters in the system has a marginal effect on reducing evacuation times." (Lines 493-500)

**Comment 6:**

- Are system wide shortest routes calculated for each of households that have yet to decide to evacuate at each time step of the simulation? This is not clear and perhaps affects the interpretation of the results regarding the superiority of centrally planned routes. What if households are just given live updates on congestion and then let them

decide on their own vs a route that is centrally planned before the flood hits. Centrally planned routes may still be better if they are repeatedly calculated at each time step of simulation where central planners also have information on congestion on various routes and it would be interesting to see how this fares compared to agents deciding on their own route but with live information on congestions. Perhaps the authors may want to provide result on this so the two cases can be fairly evaluated (self-organization for evacuation vs centrally planned one - which one is better?)

**Response to comment 6:**

Thank you very much for the excellent comments that help us to clarify the two route search methods. We would like to address the comments in turn below.

(1) Regarding the clarification of the two route search modes, mode 1 is the shortest route search method in which an agent selects the shortest route from its original location to evacuation destination in the road network. Thus, an agent's choice of route in mode 1 will not be affected by its departure time, because the agent will always choose the shortest route regardless of the time at which it starts to evacuate. In contrast, model 2 applies a global optimization method so that the agents' routes are optimized to achieve system-level evacuation efficiency. In mode 2, agents' evacuation routes will be affected by real-time traffic condition and the evacuation status of other agents. Therefore, an agent's evacuation route might be different if it starts evacuation at a different time. We have added some text in the revised manuscript for clarification, which read as follows.

"An agent's choice of evacuation route in mode 1 will not be affected by its departure time, because the agent will always choose the shortest route regardless of the time at which it starts to evacuate." (Lines 226-228)

"In contrast with mode 1, an agent's evacuation routes in mode 2 will be affected by real-time traffic condition and the evacuation status of other agents. Therefore, an agent's evacuation route in mode 2 might be different if it starts evacuation at a different time." (Lines 231-234)

(2) We thank the reviewer for proposing an alternative route search method, in which

the agents are given real-time updates on traffic congestion and let them determine evacuation routes (We may call this travel method "mode 3"). We agree with the reviewer that the global optimization route search method (mode 2) would still be better than mode 3, because mode 2 focuses on achieving system-level evacuation efficiency while mode 3 focuses on achieving individual-level evacuation efficiency. Furthermore, among the three route search methods, we hypothesize that the evacuation performance of mode 3 is between that of mode 1 and mode 2, and it would be interesting to explicitly quantify the differences among the three route search modes. Unfortunately, the functionality of mode 3 is currently not available in the latest version of MATSim software (the traffic simulation model used in this study). So we would like to leave it as a research limitation to be addressed in future work. We have added some text in the revised manuscript to discuss this research extension, which read as follows.

"Future study could improve the transportation model to consider more complex agent-agent and agent-environment interactions during evacuation processes. For instance, besides the two route search methods that have been analyzed in this study, future work may consider another type of route search method, in which agents have access to the real-time information on traffic conditions and may decide to change their evacuation routes over time. This extension will enhance the functionality of the transportation model and improve the simulation of agent behaviors during evacuation processes."
(Lines 717-724)

**References:**

Barendrecht, M.H., Viglione, A., Kreibich, H., Merz, B., Vorogushyn, S., Blöschl, G., 2019. The Value of Empirical Data for Estimating the Parameters of a Sociohydrological Flood Risk Model. Water Resour. Res. 55, 1312–1336. https://doi.org/10.1029/2018WR024128

Di Baldassarre, G., Viglione, A., Carr, G., Kuil, L., Salinas, J.L., Bloschl, G., 2013. Socio-hydrology: Conceptualising human-flood interactions. Hydrol. Earth Syst. Sci. 17, 3295–3303. https://doi.org/10.5194/hess-17-3295-2013

Du, E., Cai, X., Sun, Z., Minsker, B., 2017. Exploring the Role of Social Media and Individual Behaviors in Flood Evacuation Processes: An Agent-Based Modeling Approach. Water Resour. Res. 53, 9164–9180. https://doi.org/10.1002/2017WR021192

Fuchs, S., Karagiorgos, K., Kitikidou, K., Maris, F., Paparrizos, S., Thaler, T., 2017. Flood risk perception and adaptation capacity: A contribution to the socio-hydrology debate. Hydrol. Earth Syst. Sci. 21, 3183–3198. https://doi.org/10.5194/hess-21-3183-2017

Girons Lopez, M., Di Baldassarre, G., Seibert, J., 2017. Impact of social preparedness on flood early warning systems. Water Resour. Res. 53, 522–534. https://doi.org/10.1002/2016WR019387

Lindell, M., Sorensen, J., Baker, E., Lehman, W., 2020. Community response to hurricane threat: Estimates of household evacuation preparation time distributions. Transp. Res. Part D 85, 102457. https://doi.org/10.1016/j.trd.2020.102457

Pande, S., Sivapalan, M., 2017. Progress in socio-hydrology: a meta-analysis of challenges and opportunities. WIREs Water 4, e1193. https://doi.org/10.1002/wat2.1193

Troy, T.J., Konar, M., Srinivasan, V., Thompson, S., 2015. Moving sociohydrology forward: A synthesis across studies. Hydrol. Earth Syst. Sci. 19, 3667–3679. https://doi.org/10.5194/hess-19-3667-2015

Viglione, A., Di Baldassarre, G., Brandimarte, L., Kuil, L., Carr, G., Salinas, J.L., Scolobig, A., Blöschl, G., 2014. Insights from socio-hydrology modelling on dealing with flood risk - Roles of collective memory, risk-taking attitude and trust. J. Hydrol. 518, 71–82. https://doi.org/10.1016/j.jhydrol.2014.01.018

Yu, D.J., Haeffner, M., Jeong, H., Pande, S., Dame, J., Di Baldassarre, G., Garcia-Santos, G., Hermans, L., Muneepeerakul, R., Nardi, F., Sanderson, M.R., Tian, F., Wei, Y., Wessels, J., Sivapalan, M., 2022. On capturing human agency and methodological interdisciplinarity in socio-hydrology research. Hydrol. Sci. J.

67, 1905–1916. https://doi.org/10.1080/02626667.2022.2114836